# Multi-Objective Optimization and Test of a Tractor Drive Motor

Mengnan Liu [1,2], Yanying Li [1], Sixia Zhao [1,2], Bing Han [3], Shenghui Lei [1] and Liyou Xu [1,2,*]

1. School of Vehicle and Traffic Engineering, Henan University of Science and Technology, Luoyang 471003, China; liumengnan27@163.com (M.L.); lyy_1997@outlook.com (Y.L.); zsx19920505@163.com (S.Z.); leishenghui95@163.com (S.L.)
2. State Key Laboratory of Power System of Tractor, Luoyang 471003, China
3. College of Engineering, China Agricultural University, Beijing 100193, China; hanbingla@cau.edu.cn
* Correspondence: xlyou@haust.edu.cn

**Abstract:** The design objectives of the structural parameters of the tractor drive motor are diverse, and the constraints are complex. It is difficult to optimize the overall performance of the unit by using the empirical method and single-objective optimization method. This paper proposes a multi-objective optimization method for tractor drive motors based on an improved Non-dominated Sorting Genetic Algorithm II (NSGA-II). Constraints are formulated according to the inherent characteristics of the motor itself and the characteristics of the tractor's working conditions. The objective function was established with the heat loss of the drive motor and the total efficiency of the drive system. Based on the designed solution process of NSGA-II algorithm, an example optimization was *carried* out, and the tractor electromechanical drive system was carried out with the single-objective optimization results of the optimal energy use efficiency of the drive motor and the optimal mechanical transmission efficiency of the transmission system as the control group. The test results show that compared with the control group, the proposed multi-objective optimization method can make the overall tractor system efficiency the highest, and the maximum and rated values of the total efficiency $\eta_q$ of the drive system of the multi-objective optimization design scheme. Compared with the optimal design scheme with $\eta_{me}$ as a single objective, it was increased by 2% and 1.4%, respectively, and compared with the optimal design scheme with $\eta_{tr}$ as a single objective, it is improved by 26.5% and 73.6%, respectively. It can provide an effective calculation method for the motor design problem in the subsequent development of the tractor electromechanical drive system.

**Keywords:** electric drive tractor; drive motor; multi-objective optimization; bench test

## 1. Introduction

Electric tractors have the advantages of zero-emission, no pollution, high energy efficiency, and low noise, which is the direction of the future development and transformation of agricultural power machinery. Tractors with electric motors as the main power unit can simultaneously match a variety of power sources, such as batteries, power grids, generator sets, and other power sources, which can better deal with the shortage of petroleum resources in the future and improve air pollution problems caused by mechanized agricultural production [1,2]. Due to the large number of types of power generation energy, if the proportion of clean energy in the power system is high, the wide application of electric tractors will significantly reduce carbon emissions, which is conducive to the sustainable development of agricultural environment. Moreover, compared with power shifting systems and mechanical hydraulic continuously variable transmissions, drive motors have higher energy efficiency and mechanical characteristics of continuously variable transmission, with a larger starting torque, better traction, and acceleration performance [3,4]. Through the matching of two to three mechanical gears, the tractor can meet all the speed requirements in crawling, plowing, and transportation operations at high operating efficiency [5].



Usually, the design and selection of the drive motors have a significant influence on the overall efficiency of the whole vehicle owing to the significant differences in the working conditions, structure, and demand characteristics of different types of electric vehicles. For example, the mechanical transmission system of general electric vehicles is relatively simple because of a low torque reserve requirement and the absence of a full-load output requirement at low speed. Therefore, most studies believe that selecting a high-speed drive motor can effectively take advantage of its lower copper loss and improve system efficiency [6]. For some special vehicles with stable speed or vehicles driven by in-wheel motors, due to the small total speed regulation, related research considers that the selection of direct-drive motors with low speed and large torque can effectively reduce the complexity of mechanical transmission systems and improve the system efficiency [7,8]. However, under low-speed operating conditions, such as plowing, the speed of these conditions is usually 5–10 km/h, which has the requirement of full tractor power output; and the maximum speed exceeds 50 km/h, resulting in the total speed ratio of the tractor reaching 5–10. The speed regulation ratio of the general vehicle drive motor is only 2–3.5, which results in more mechanical gears and a larger transmission ratio for matching the transmission system when designing the tractor drive motor [8]. Overly complicated mechanical transmission devices will reduce the mechanical transmission efficiency of the tractor and cannot take advantage of the high efficiency of high-speed motors. Thus, the existing design and optimization methods for other types of electric vehicle drive motors are not suitable for tractors' drive motors [9].

The author of this paper has conducted some research in the field of related electric tractors. Given the large randomness of the load peak power and high-frequency power under the heavy-duty operation mode of the tractor, a 18.5 kW electric tractor using a DC–DC converter in parallel with a supercapacitor is proposed. Research on composite power supply schemes, etc., provides a relevant basis for the study of tractor drive motor optimization [10]. For this reason, the tractor drive system, including the drive motor and the mechanical transmission system, needs to be taken as the research object, with the lowest heat loss of the drive motor body and the highest overall energy use efficiency of the drive system as the optimization goals. Relevant constraints are established based on tractor operating conditions and characteristic requirements, and multi-objective optimization of the main parameters of the drive motor is performed to improve the energy use efficiency of the electric drive tractor system and optimize the thermal balance of the motor itself. In addition, it can also reduce the cost of the power consumption of electric tractor operations, improve the overall performance of the operating unit, and improve the economic benefits of farmers.

In recent years, the field of intelligent algorithm technology has developed rapidly, providing an effective method for solving multi-objective optimization problems. This essay proposes a multi-objective optimization method for tractor drive motor on the basis of the non-dominated sorting genetic algorithm II (NSGA-II).

## 2. Optimization Parameters of Drive Motor

The technical advantages of brushless DC motors (BLDC) over DC motors and induction motors are as follows: (1) Through electronic control commutation, the operation characteristics are similar to those of the DC motor, with good controllability and a wide speed regulation range. (2) The drive motor is controlled by rotor position feedback control signal and electronic multi-phase inverter. (3) The absence of a brush and commutator results in high reliability. (4) The air gap magnetic field generated by a permanent magnet results in a high power factor. (5) The energy use efficiency of BLDC motors can reach more than 92.5%, which can increase by more than 12% compared with induction motors [11]. In this context, the BLDC motor is selected as the drive motor of the electric tractor, and the multi-objective optimization design method of parameters is studied.

The modeling setting conditions are as follows: (1) The transition process of current change is ignored. (2) The effect of armature reaction on the air gap magnetic field is not

considered. (3) The rotor-induced current effect is neglected. (4) The power switch is replaced by equivalent resistance. (5) The rotor speed fluctuation is not taken into account.

Figure 1 shows the equivalent circuit model of the BLDC motor [12] at a single state angle $\theta_x$. The voltage balance equations are:

$$U_m = U_{DO} - N_c \Delta U_m,\tag{1}$$

$$R_{eq} = R_m + N_c R_g,\tag{2}$$

$$U_m = E_{eq} + i_{eq} R_{eq},\tag{3}$$

where $U_m$ is the drive motor voltage, V; $N_c$ is the number of series switches; $\Delta U_m$ is a single constant tube voltage drop, V; $U_{eq}$ is the bus voltage, V; $R_{eq}$ is the equivalent resistance of the drive motor, Ω; $R_g$ is the equivalent resistance of the motor power bridge, Ω; $R_m$ is the internal resistance of the motor, Ω; $i_{eq}$ is the instantaneous current, A; and $E_{eq}$ is the back electromotive force (BEMF) of the equivalent circuit, V.

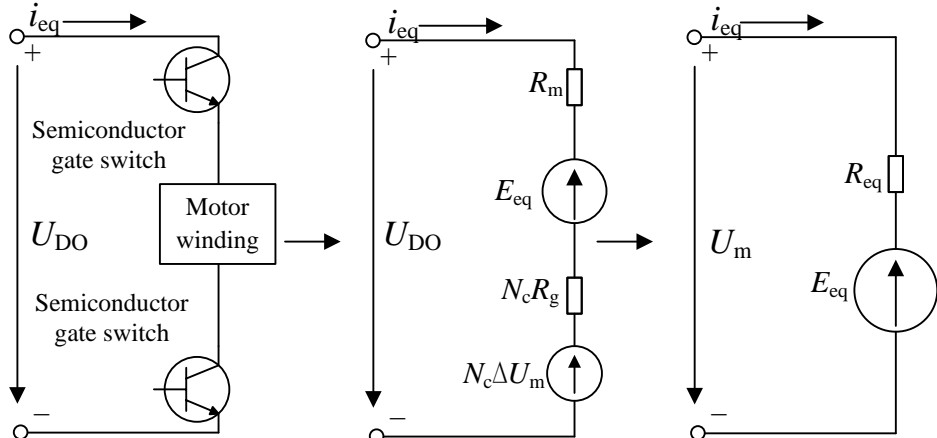

**Figure 1.** Equivalent circuit of BLDC motor under single electric angle.

The BEMF of the equivalent circuit can be expressed as follows:

$$E_{eq} = E_m f(\theta_x),\tag{4}$$

$$E_m = k_{eq} \omega_m,\tag{5}$$

$$E_{eq} i_{eq} = T_e \omega_m,\tag{6}$$

where $E_m$ is the amplitude of BEMF, V; $k_{eq}$ is the equivalent winding BEMF coefficient, V·min/r; and $\theta_x$ is the state angle of the rotor position; $f(\theta_x)$ is the waveform function of $\theta_x$; $\omega_m$ is the speed of the driving motor, r/min.

In accordance with Equations (3)–(6), the transient and average values of the electromagnetic torque are derived as follows:

$$T_e = \frac{U_m k_{eq} f(\theta) - k_{eq}^2 f^2(\theta) \omega_m}{R_{eq}},\tag{7}$$

$$T_{av} = \frac{1}{\theta_x} \int_{\theta_1}^{\theta_2} T_e d\theta,\tag{8}$$

where $T_{av}$ is the average value of electromagnetic torque in the state angle, N·m; $\theta_1$ and $\theta_2$ are the start and stop thresholds of the state angle.

$k_{av}$ is the average coefficient of the BEMF waveform, $k_{ef}$ is the effective coefficient of the BEMF waveform [13], and variable substitution is performed, which is shown in the following:

$$k_{av} = \frac{1}{\theta_x} \int_{\theta_1}^{\theta_2} f(\theta)d\theta, \tag{9}$$

$$k_{ef} = \sqrt{\frac{1}{\theta_x} \int_{\theta_1}^{\theta_2} f^2(\theta)d\theta} \tag{10}$$

Substituting Equations (8)–(10) into Equation (7), as follows:

$$T_{av} = \frac{U_m k_{eq} k_{av} - k_{eq}^2 k_{ef}^2 \omega_m}{R_{eq}}. \tag{11}$$

When $\omega_m \approx 0$, the working torque of the drive motor reaches the maximum value, substituting into Equation (11), which is described as follows:

$$T_{max} = k_{eq} k_{av} \frac{U_m}{R_{eq}} = k_T i_{max}, \tag{12}$$

$$k_T = k_{eq} k_{av}, \tag{13}$$

where $T_{max}$ is the maximum torque, N·m; $i_{max}$ is the current at the maximum torque, A; $k_T$ is the motor torque coefficient, N·m/A.

When the drive motor is working without a load, $T_{av} \approx 0$; the speed of the BLDC motor is $\omega_m \approx \omega_{max}$, and the following equations are valid:

$$\omega_{max} = \frac{U_m}{k_E}, \tag{14}$$

$$k_E = \frac{k_{ef}^2}{k_{av}} k_{eq}, \tag{15}$$

where $k_E$ is the coefficient of BEMF, V·min/r.

Through replacement with the coefficient of BEMF and the coefficient of torque and the substitution of Equations (13) and (15) into Equation (11), $T_{av}$ can be calculated as follows:

$$T_{av} = \frac{k_T U_m}{R_{eq}} - \frac{k_T k_E U_m}{R_{eq}} \omega_m. \tag{16}$$

With $D_m$ defined as the viscous damping coefficient, the functional relationship between $T_{max}$, $T_m$, $\omega_{max}$ and $\omega_m$ can be established as:

$$D_m = \frac{k_E k_T}{R_{eq}} = \frac{T_{max}}{\omega_{max}}, \tag{17}$$

$$T_m = T_{max} - D_m \omega_m, \tag{18}$$

$$\omega_m = \omega_{max} - \frac{T_m}{D_m}. \tag{19}$$

The motor stator winding current in $\theta_x$ is described as:

$$i_m = \frac{1}{\theta_x} \int_{\theta_1}^{\theta_2} i_{eq}d\theta = i_{max} - \frac{k_T}{R_{eq}} \omega_m. \tag{20}$$

In the three-phase state, $k_E \approx k_T$ can be considered approximately, with Equation (14) substituted into Equation (19), the relationship between speed, torque and voltage of the BLDC motor is obtained as follows:

$$\omega_m = \frac{U_m}{k_E} - \frac{T_m}{D_m}.$$

(21)

When the electric tractor works, the voltage of the drive motor is mainly restricted by the speed regulation characteristics of the load terminal. In accordance with Equations (16) and (20), the main variables that affect the mechanical characteristics of the motor are $k_E$, $k_T$, and $D_m(k_E, k_T, R_{eq})$.

## 3. Design Objective Function of Drive Motor

In accordance with Equation (20), the stator current of the BLDC motor has a linear inverse relationship with the mechanical speed. As a matter of fact, the load characteristic of an electric tractor has the characteristics of low speed and high resistance under farmland operating conditions. The ratio of the output torque of the drive motor to the maximum torque is large, and $\omega_m/\omega_{max}$ is low, which will result in a large current of the drive motor [14]. The energy loss of BLDC includes copper loss, iron loss, and mechanical loss. Except for some iron loss, the remaining energy loss is all heat loss. Under the condition of a high current, the copper loss of the drive motor is large, thereby increasing the heat generation of the drive motor and the working load of the radiator. As a result, the reliability and working life are reduced [15].

### 3.1. Heat Loss Objective Function

With the mechanical friction loss being not taken into consideration, the heat loss rate $\eta_{mA}$ of BLDC motor can be mainly related to $R_{eq}$ and $i_m$. The formula is as follows:

$$\eta_{mA} = \frac{R_{eq}i_m^2}{U_m i_m}.$$

(22)

Introduce Equations (14) and (21) into $\eta_{mA}$, which is:

$$\eta_{mA} = \frac{T_m R_{eq}}{\omega_{max}k_T^2}$$

(23)

In Equation (23), the heat loss of the BLDC motor is mainly affected by motor torque coefficient, winding resistance, no-load speed, and mechanical torque. Therefore, under the same output torque, the drive motor with a larger no-load speed can reduce the system heat loss and reduce the heating phenomenon. With Equation (17) integrated into Equation (23), taking $k_E \approx k_T$, the process can be described as:

$$\eta_{mA} = \frac{T_m}{\omega_{max}D_m}$$

(24)

In addition, with the insertion of Equation (19) into Equation (23), which is shown in the following:

$$\eta_{mA} = \frac{T_m^2}{T_m^2 + P_m D_m}$$

(25)

where $P_m$ is the power of the driving motor, kW.

As a result of the operation of the electric tractor, the drive motor is mainly output at the rated working condition, taking $T_m = T_{mT}$; $P_m = P_{mT}$. When the rated power of the

tractor is determined, $T_m$ and $D_m$ are used as the main variables in the objective function of heat loss. The calculation formula is as follows:

$$f_{\text{Amt}}(D_m, T_{mT})_{\text{best}} = \min \eta_{mA}(D_m, T_{mT}) = \frac{T_{mT}^2}{T_{mT}^2 + P_{mT}D_m} \tag{26}$$

where $T_{mT}$ is the rated torque of the drive motor, N•m.

*3.2. Total Efficiency Function of the Drive System*

Improving the energy efficiency of the drive motor can improve the economy of the electric tractor. The following formula can be calculated as:

$$\eta_{\text{me}} = 1 - \eta_{mA} \tag{27}$$

$$\eta_{\text{meT}} = 1 - \frac{T_{mT}^2}{T_{mT}^2 + P_{mT}D_m} \tag{28}$$

where $\eta_{\text{me}}$ is the energy efficiency of the motor, %; and $\eta_{\text{meT}}$ is the energy efficiency of the rated operating point of the motor, %.

Based on Equations (26) and (28), $\eta_{\text{me}}$ increases with the rated power of the motor when the rated torque and hardness are the same; the $\eta_{\text{me}}$ of the BLDC motor with a high rated speed is higher when the rated power is the same.

As a result of the large load torque of the electric tractor, the total transmission ratio of the transmission system is relatively large when the high-speed motor is used, the transmission stages increase in number, and the transmission efficiency is lower. Therefore, an objective function is established, with the highest total efficiency of the electric tractor drive system taken as the design goal.

The transmission efficiency of a single-stage gear pair is 0.98. Then, the total efficiency of the drive system is described as:

$$\eta_q = \eta_{\text{meT}}\eta_{\text{tr}} = (1 - \frac{T_{mT}}{D_m\omega_{\max}})0.98^{\tau_1} \tag{29}$$

According to the [16], referring to the previous design experience, the threshold range is set as:

$$\begin{cases} \tau_1 = 1 & 0 \leq \varsigma_1 \leq 1 \\ \tau_1 = 2 & 1 \leq \varsigma_1 \leq 7.1 \\ \tau_1 = 3 & 7.1 \leq \varsigma_1 \leq 28 \\ \tau_1 = 4 & 28 \leq \varsigma_1 \leq 31.5 \end{cases} \tag{30}$$

$$\varsigma_1 = \frac{F_T r_q}{5T_{mT}} \tag{31}$$

where $\varsigma_1$ is the design factor of the total transmission ratio; $\tau_1$ is the design value of the transmission stages; $\eta_{\text{tr}}$ is the total efficiency of the transmission system, %; and $\eta_q$ is the total efficiency of the drive system, %.

The total efficiency objective function of the drive system is as follows:

$$g_{\text{emt}}(D_m, T_{mT}, \omega_{\max})_{\text{best}} = \max \eta_{\text{meT}}(D_m, T_{mT}, \omega_{\max})\eta_{\text{tr}}(T_{mT}) = (1 - \frac{T_{mT}}{D_m\omega_{\max}})0.98^{\tau_1(T_{mT})} \tag{32}$$

## 4. Design Constraints of Drive Motor

The constraint function is derived based on the stator slot full slot rate limit, the field-weakening speed regulation capability limit, the torque reserve limit, and the rated power limit.

### 4.1. Maximum Slot Full Rate Constraints

The full slot ratio of the drive motor is the ratio of the winding volume of each slot of the stator to the single slot volume. If the stator winding full slot ratio is too high, then the winding coil distribution will be too compact, and the performance will be reduced. The maximum slot full rate constraints are:

$$W_{\mathrm{p}}(D_{\mathrm{m}}) = \frac{k_{\mathrm{E}}}{\frac{k_{\mathrm{ef}}^2}{k_{\mathrm{av}}} k_{\mathrm{e}} k_{\mathrm{w}} D_{\mathrm{a}}(D_{\mathrm{m}}) B_{\mathrm{m}}(D_{\mathrm{m}}) L_{\mathrm{st}}(D_{\mathrm{m}})} \qquad (33)$$

$$N_{\mathrm{cu}}(D_{\mathrm{m}}) = \frac{W_{\mathrm{p}}(D_{\mathrm{m}})}{4} \qquad (34)$$

$$k_{\mathrm{s}}(D_{\mathrm{m}}) = \frac{2 a_{\mathrm{cu}} d_{\mathrm{cu}}^2 N_{\mathrm{cu}}(D_{\mathrm{m}})}{A_{\mathrm{s}}} < k_{\mathrm{ss}} \qquad (35)$$

where $D_{\mathrm{a}}$ is the diameter of the stator core, cm; $B_{\mathrm{m}}$ is the amplitude of the air gap magnetic flux density, T; $L_{\mathrm{st}}$ is the length of the stator core, cm; $k_{\mathrm{w}}$ is the winding factor; $k_{\mathrm{e}}$ is the ratio of $k_{\mathrm{ep}}$ and single-phase winding-back EMF coefficient; $W_{\mathrm{p}}$ is the number of winding turns; $a_{\mathrm{cu}}$ is the number of parallel shares; $N_{\mathrm{cu}}$ is the number of single-coil turns; $A_{\mathrm{s}}$ is the area of the single stator slot, mm$^2$; and $k_{\mathrm{s}}$ is the full slot rate; $k_{\mathrm{ss}}$ is the maximum slot full rate.

### 4.2. Field-Weakening Speed Regulation Constraints

The electric tractor drive motor adjusts the speed by using Equation (21). Through voltage regulation and field-weakening speed regulation, the drive motor keeps running at a constant torque before the rotation speed is reached $\omega_{\mathrm{mT}}$ and maintains a constant power output after the rotation speed reaches $\omega_{\mathrm{mT}}$. The speed ratio $k_\omega = \omega_{\max}/\omega_{\mathrm{mT}}$ is defined to characterize the motor speed regulation performance. BLDC motors have permanent magnet rotors. The field-weakening speed regulation capability is limited by the inherent magnetic field of the permanent magnets. The air gap magnetic field can be weakened only by the stator magnetic field opposite the rotor magnetic field. The field-weakening speed regulation constraint is:

$$k_\omega = \frac{\omega_{\max}}{\omega_{\mathrm{mT}}} < 2 \qquad (36)$$

The mechanical characteristics of BLDC motors are determined by hardness. The hardness and viscosity damping coefficients are numerically equal. In accordance with Equation (36), $k_\omega = 2$ is taken and the relationship between the speed regulation characteristics and inherent characteristics of the drive motor is established as:

$$D_{\mathrm{m}} = \frac{\Delta T_{\mathrm{m}}}{\Delta \omega_{\mathrm{m}}} \approx \frac{T_{\mathrm{mT}}}{\omega_{\max} - \omega_{\mathrm{mT}}} \qquad (37)$$

$$T_{\mathrm{mT}} = D_{\mathrm{m}} \omega_{\mathrm{mT}} | k_\omega = 2 \qquad (38)$$

### 4.3. Constraints of Torque Reserve Coefficient

The rated torque of the BLDC motor is limited by the rated traction of the electric tractor. On the basis of the resistance change factors caused by working conditions and the performance of agricultural tools, the torque reserve coefficient is 10–20% [17]. The equation is as follows:

$$F_{\mathrm{TN}} = (1.1 \sim 1.2) F_{\mathrm{Ta}} \qquad (39)$$

In Equation (39), $F_{\mathrm{TN}}$ is the rated traction force of the electric tractor, N; and $F_{\mathrm{Ta}}$ is the plow resistance under the working condition, N.

$$F_{\mathrm{Ta}} = \frac{T_{\mathrm{mT}} i_{\mathrm{ky}} i_0 \eta_0 \eta_{\mathrm{ky}}}{r_{\mathrm{q}}} \qquad (40)$$

In Equation (40), $i_{ky}$ is the Y gear transmission ratio of the transmission; $i_0$ is the transmission ratio of the central drive-train; $\eta_{ky}$ is the transmission efficiency of the transmission, %; $\eta_0$ is the transmission efficiency of the central drive-train, %; and $r_q$ is the driving radius, m.

Equations (29) and (39) are substituted into Equation (40) to obtain:

$$k_{TN} = 5\frac{\varsigma_1 T_{mT} 0.98^{\tau_1}}{r_q F_{Ta}} \tag{41}$$

$$1.1 \leq k_{TN} \leq 1.2 \tag{42}$$

where $k_{TN}$ is the torque reserve factor of the drive motor, N.

*4.4. Tractor Rated Power Constraints*

The drive motor rated power needs to meet the requirements of the electric tractor rated power. Because the motor speed regulation process is a slope change around the rated point, there is $D_m = T_{mT}/\omega_{mT}$. In accordance with Equations (20), (27), and (42), the relationship between the BLDC motor viscous damping coefficient and the rated power of the electric tractor is derived as:

$$P_{TN} \leq \frac{T_{mT}^2}{9550 D_m} \tag{43}$$

With the use of Equation (43), when the rated torque or no-load speed is determined, a nonlinear inverse relationship is found between the mechanical hardness of the electric tractor drive motor and the rated power.

## 5. Optimization Algorithm and Example Design
*5.1. Optimization Algorithm Design*

The optimization algorithms that are used to deal with multi-objective problems include the intensity Pareto evolutionary algorithm, particle swarm optimization (PSO) algorithm, non-dominated genetic algorithm, and improved non-dominated genetic algorithm. Compared with the intensity Pareto evolutionary algorithm, the NSGA-II algorithm has better convergence, front-end distribution, and diversity occurrence mechanism [18]. The NSGA-II algorithm has better diversity than PSO [19,20], and it more greatly reduces the complexity of the algorithm compared with the non-dominated genetic algorithm by adding an elite strategy, density value estimation strategy, and fast unsupported sorting strategy [21,22]. The NSGA-II algorithm has been widely used in engineering problems such as power grid system planning, vehicle path optimization, and hybrid vehicle drive system matching [23–25].

The NSGA-II algorithm process is as follows:

Firstly, a population of N individuals is randomly generated, the parent population, and the individuals in the parent population are non-dominantly sorted. Secondly, the individual crowding degree is calculated, the individual crowding degree determines the level, and the appropriate individual is selected by selecting the operator to cross, mutate and other operations to generate a new next-generation population. Finally, the elite strategy operation is carried out, and the obtained N individuals are used as the new parent to continue the above iterative process until the optimization accuracy reaches the termination condition. As shown in Figure 2, the software implementation process of the NSGA-II. algorithm is used to process the optimization model of the tractor transport unit using the gamultiobj function based on the NSGA-II algorithm; the gacommon function is called to determine the constraint type of the optimization model; the population evolution is initialized by the steamgammultiobj function; and the optimal value of the target function is solved by calling the gumultiobjsolve function.

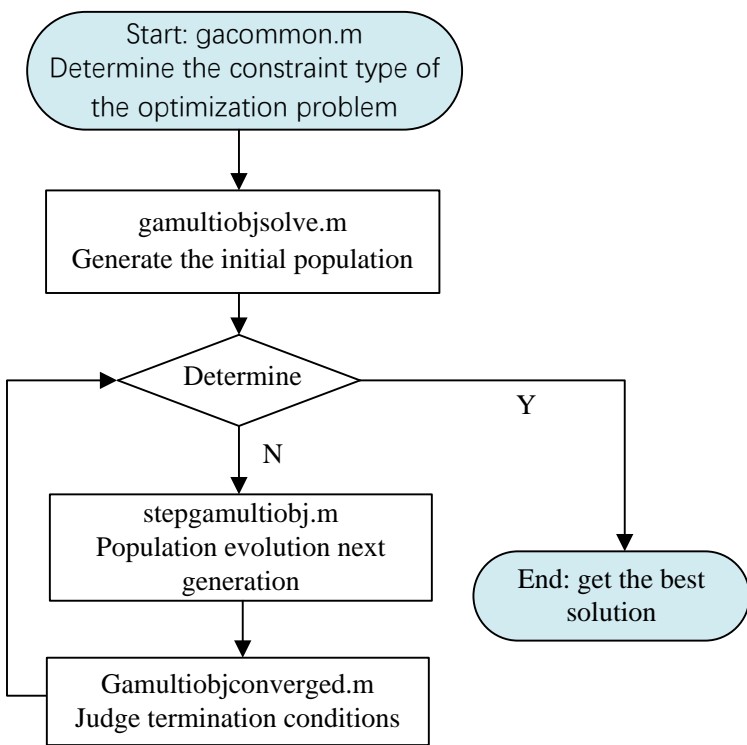

**Figure 2.** Software implementation of NSGA-II algorithm.

The 36.8 kW electric tractor developed in the previous period is taken as the research object to optimize the design of the drive motor. The overall parameters and dynamic performance parameters are known from the literature [26].

Figure 3 shows the optimal design algorithm of an electric tractor drive motor based on the NSGA-II algorithm. As seen in the figure, the design constraints of $T_{mT}$ are established by inputting the parameters of the electric tractor, such as gravity, rated traction force, and centroid position, into the motor reserve torque constraint Equations (39)–(42). The manual and product catalog is checked, the basic parameters of the winding are determined, the slot full rate constraint Equation (35) is imported, and the design constraint of $D_m$ is established. The design constraints of $T_{mT}$ and $D_m$ are imported into Equation (43) to establish the $P_{TN}$ constraint. The relationship between the rated parameters of the drive motor is:

$$P_{TN} = \frac{T_{mT}\omega_{mT}}{9550} \tag{44}$$

The design constraints of $P_{TN}$ and $T_{mT}$ are introduced into Equation (44) to obtain the constraint equation of $\omega_{mT}$, and the constraint of the motor speed regulating ability is introduced into Equation (36). Then, the design constraint of $\omega_{max}$ is established. From the Sheffield University MATLAB GA toolbox, the two functions that are gacommon.m and gamultiobjsolve could easily be called [27]. Through gacommon.m processing constraint types, the boundary conditions are established. The global solution of heat loss objective Equation (26) and transmission efficiency objective Equation (32) is carried out by gamultiobjsolve. Afterward, the fitness function deviation is determined, and the individual front-end distribution of the solution set of $D_m$ and $T_{mT}$ is solved; the optimal solution is selected as the design value of traction motor parameters. The drive motor $\omega_{max}$ is calculated according to Equation (21). Finally, in accordance with the electric tractor's bus voltage of the drive motor is set, and the other intrinsic parameters of the drive motor are calculated by Equations (14) and (21).

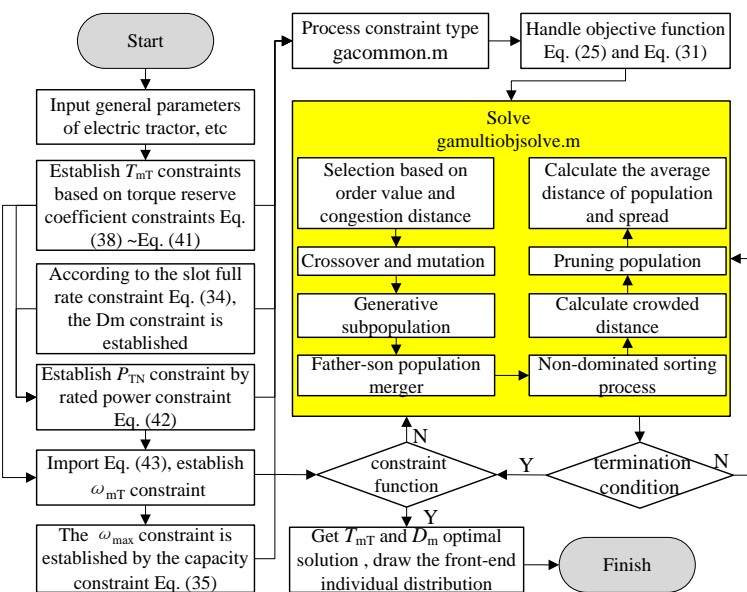

**Figure 3.** Optimization algorithm of the drive motor.

*5.2. Optimization Example*

The optimal front-end individual coefficient of the NSGA-II algorithm is set to 0.3, the population size is 200, the maximum evolutionary algebra is 500, the stop algebra is 500, and the fitness function value deviation is $1 \times 10^{-1000}$.

Figure 4 shows the individual front-end distribution of the parameter set of the drive motor of the electric tractor. On the basis of Equation (29) and objective function Equation (32), when the value of τ1 is the same, the total efficiency of the electric tractor drive system is the same. From Equation (30) $g_{tx}(D_m, T_{mT}, \omega_{max})$ is established, and $g_{emt}(D_m, T_{mT}, \omega_{max})$ is substituted. The figure shows that when τ1 = 1, the heat loss of the drive motor is high; when τ1 = 2, $f_{Amt}(D_m, T_{mT})$ and $g_{tx}(D_m, T_{mT}, \omega_{max})$ has a nonlinear inverse relationship. In the solution set of τ1 = 2, a large $g_{tx}(D_m, T_{mT}, \omega_{max})$ corresponds to a slight heat loss. The design values of the motor parameters are calculated by $D_m$, $T_{mT}$, and $\omega_{max}$ corresponding to the individual point C.

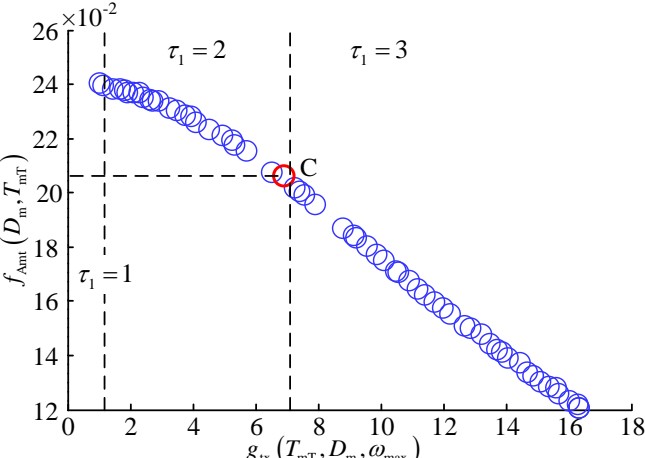

**Figure 4.** Pareto front-end individual distribution of objective.

Table 1 shows the comparison of parameter design results of electric tractor drive motors. The single-objective optimal design scheme of $\eta_{me}$ optimal and $\eta_{tr}$ optimal was used as the control group.

**Table 1.** Drive motor parameters are designed by different methods.

| Design Method | Multi-Objective Optimization | Optimization Aimed at $\eta_{\mathrm{me}}$ | Optimization Aimed at $\eta_{\mathrm{tr}}$ |
|---|---|---|---|
| $P_{\mathrm{mT}}/\mathrm{kW}$ | 36.8 | 36.8 | 36.8 |
| $T_{\mathrm{mT}}/\mathrm{N\cdot m}$ | 216 | 92.5 | 1445 |
| $\omega_{\mathrm{mT}}/\mathrm{r\cdot min}^{-1}$ | 1627 | 3800 | 243 |
| $\omega_{\mathrm{max}}/\mathrm{r\cdot min}^{-1}$ | 2861 | 7500 | 496 |
| $k_{\mathrm{E}}/\mathrm{V\cdot min\cdot r}^{-1}$ | 0.133 | 0.051 | 0.766 |
| $D_{\mathrm{m}}/\mathrm{N\cdot m\cdot min\cdot r}^{-1}$ | 0.175 | 0.024 | 5.713 |
| $V_{\mathrm{mT}}/\mathrm{V}$ | 380 | 380 | 380 |

The drive motor rated power and rated voltage are the same for the same electric tractor. The table shows that in the design scheme with $\eta_{\mathrm{me}}$ as the target, the no-load speed is 7500 r/min, which belongs to the high-speed motor; in the design scheme with $\eta_{\mathrm{tr}}$ as the target, the rated torque is 1445 N·m, which belongs to low-speed high torque motor. Under the three schemes, the values of $\omega_{\mathrm{max}}/\omega_{\mathrm{mT}}$ are 1.76, 1.97, and 2, respectively, which is in line with the BLDC motor speed regulation performance constraint Equation (36).

## 6. Control Test

### 6.1. Test Design

In accordance with the calculated values of the drive motor parameters in Table 1, a similar BLDC motor is selected to conduct the loading test of the electric tractor drive system. The BLDC motor with rated speeds of 1600, 3800, and 240 r/min produced by Taian Sunshine Power Company Ltd. in the Henan Province Key Laboratory of Vehicle Energy Saving and New Energy is taken as the test object. The test equipment adopts the developed 90 kW electric tractor drive system loading test bench, which mainly includes the DJ4000-XN DL90 motor assembly dynamometer and the two-way battery simulator produced by Xi'an Xunpai Quick Charge Technology Company Ltd. With the use of the constant power measurement mode, an average of 200 test torques are inserted before the torque of three groups of motors reaches $T_{\mathrm{max}}$. Then, the upper computer reads the $\eta_{\mathrm{me}}$ and $\eta_{\mathrm{tr}}$ measured at each test torque.

During the test, WT333 power analyzer, ET4100 measurement and control instrument, and ET4300-AI data acquisition module are used to collect and process the input electrical power signal and output mechanical power signal, respectively. Table 2 lists the main parameters of the hardware equipment. The communication interfaces are as shown in Figure 5, in which the two-way battery simulator on the tested motor side and the two-way variable frequency control cabinet on the loaded motor side are both powered by the power grid; the torque and speed of the BLDC motor are measured by a HDT05 torque-speed integrated sensor, and the mechanical power of the BLDC motor is obtained by data processing. The load torque and speed are closed-loop controlled by the digital segmented PID method through the ET4100 measurement and control instrument, and the input electric power signals at both ends of the motor controller are measured by the WT333 Power Analyzer to obtain the energy efficiency of the drive motor. With the use of the ET4100 measurement and control instrument, the load torque and speed are closed-loop controlled by the digital segmented PID method; the input electric power signals at both ends of the motor controller are measured by the WT333 Power Analyzer, and the energy use efficiency of the drive motor is obtained. $\eta_{\mathrm{q}}$ is calculated by using Equations (28)–(30). Then, regression analysis on each group's corrected discrete measurement data is performed using the least squares method to obtain continuous test control results.

**Table 2.** Hardware device parameters.

| Equipment | Parameter | Numeric/Type | Production Enterprises |
|---|---|---|---|
| Two-way battery simulator | Hardware technology<br>Input power/V<br>Output power/kW<br>Output voltage/V<br>Output current/A<br>Detects the resolution<br>Mode of communication | High-power IGBT technology<br>380 (Three-phase five-wire AC)<br>100<br>670<br>−200~200<br>15 mA or 15 mV<br>CAN, Ethernet | Xi'an Xunpai |
| HDT05 torque speed integrated sensor | Rated torque/(N·m)<br>Rated speed/(r/min)<br>Precision/% | 2000<br>4000<br>±0.2 | YB2~2000 |
| WT333 power analyzer | channel<br><br>Precision/% | WT333,<br>Three-channel<br>0.2 | Yokogawa, Japan |

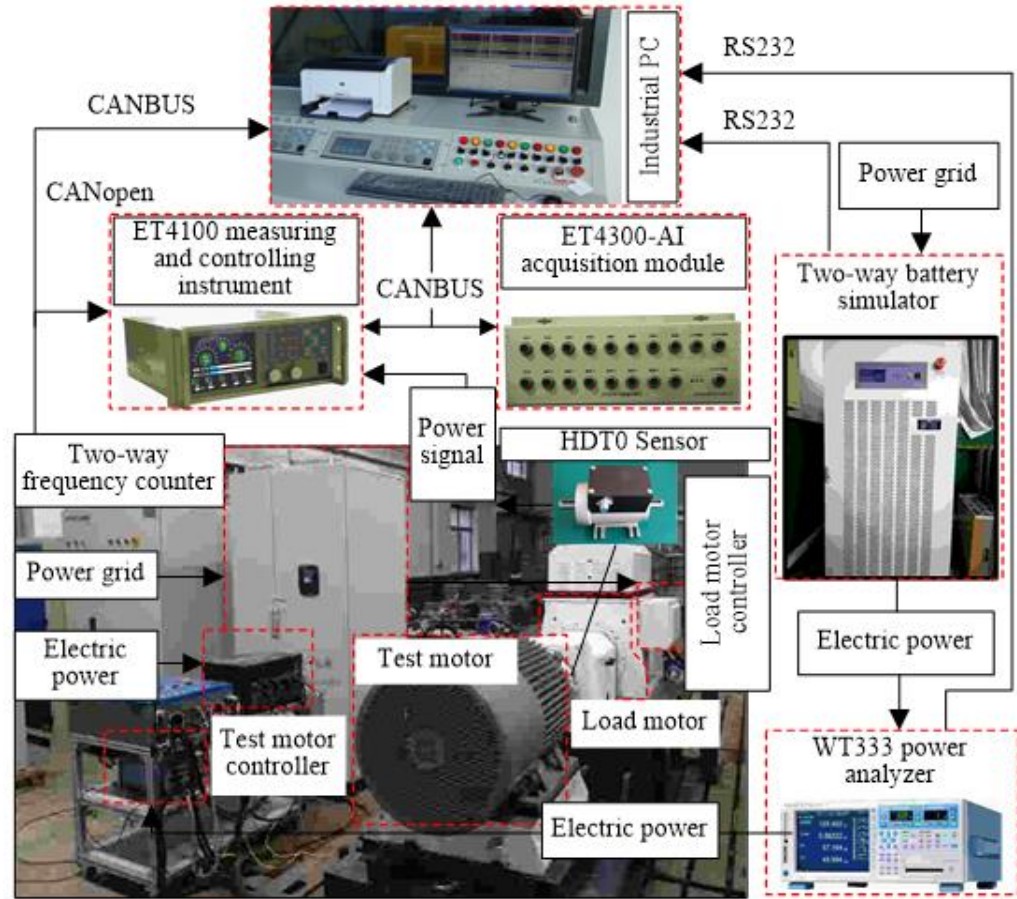

**Figure 5.** Measurement experiment of the drive system.

*6.2. Comparative Analysis*

Figure 6 shows the loading test results of the different optimization results. As can be seen from the figure, in Equations (25) and (27), $\eta_{\mathrm{me}}$ can characterize the heat loss rate. The mechanical power at the rated torque point of each scheme is equal to 36.8 kW. First, with $\eta_{\mathrm{me}}$ as the target design, the maximum values of drive motors $\eta_{\mathrm{me}}$ and $\eta_{\mathrm{q}}$ are 94.5% and 88%, respectively; $\eta_{\mathrm{me}}$ and $\eta_{\mathrm{q}}$ of the rated torque point are 94% and 86.2%, respectively. Then, with $\eta_{\mathrm{tr}}$ as the target design, the maximum values of the drive motors $\eta_{\mathrm{me}}$ and $\eta_{\mathrm{q}}$ are 69% and 63.5%, respectively; the rated torque points $\eta_{\mathrm{me}}$ and $\eta_{\mathrm{q}}$ are 15%

and 14%, respectively. With the use of the multi-objective optimization design scheme, the maximum value of drive motors $\eta_{me}$ and $\eta_q$ are 93.7% and 90%, respectively; rated torque point $\eta_{me}$ and $\eta_q$ are 88.5% and 87.6%, respectively. By comparison, the electric tractor motor designed only for the purpose of improving transmission efficiency exhibits reduced performance due to $i_m \approx T_m/k_T$; the heat loss increases because of a quadratic polynomial function relationship with the increase in the rated torque, which exceeds the transmission efficiency increase speed of Equation (28). Not only is $\eta_q$ reduced, but a large heat loss also occurs, resulting in serious heat generation of the motor and reduced reliability. The electric tractor motor designed to improve the motor energy efficiency of the drive motor has the highest maximum and rated values of $\eta_{me}$, the lowest heat loss rate, and the smallest heat generation. The rated power of the tractor is proportional to the rated traction force; therefore, the maximum speed is relatively stable. Low-power electric tractors have a relatively low rated traction force, and increasing the motor's rated torque is not obvious for the improvement of transmission efficiency, which is why a high-speed motor is used as the drive motor. High-power tractors have a highly rated traction force, a complicated transmission system, and low efficiency.

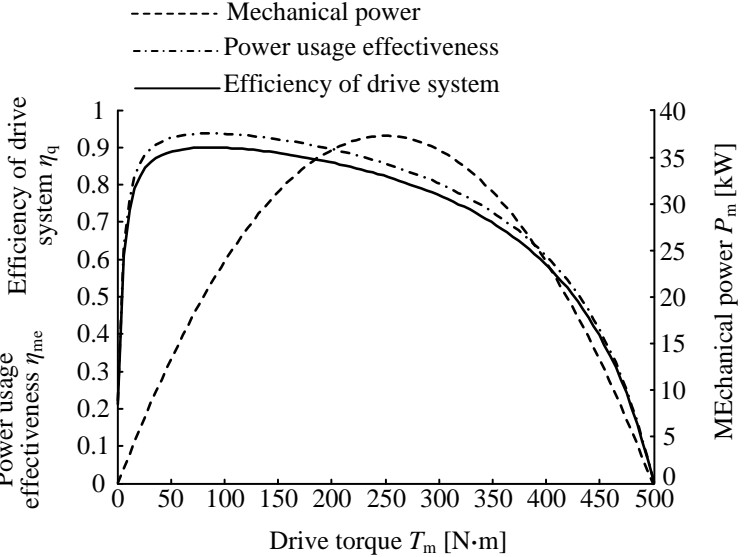

(**a**) Test result of multi-objective optimization scheme.

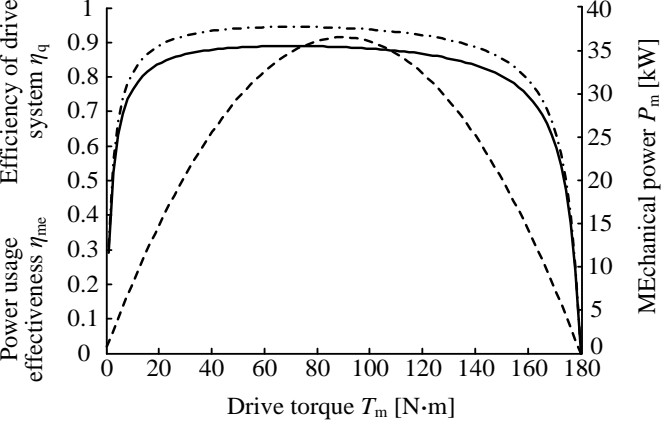

(**b**) Test result of optimization scheme aimed at η<sub>me</sub>.

**Figure 6.** *Cont.*

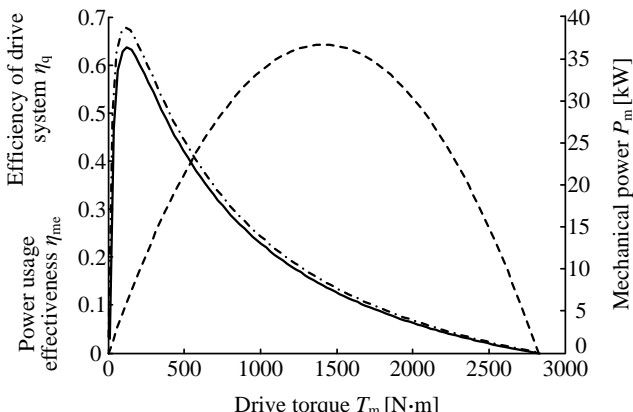

(**c**) Test result of optimization scheme aimed at $\eta_{tr}$

**Figure 6.** Test result of drive system experiment.

A tractor motor designed only to improve the energy efficiency of the drive motor results in excessively low-rated torque, thus reducing the overall efficiency of the drive system. The rated power of the research object in this paper is 36.8 kW; the maximum and rated values of the total efficiency $\eta_q$ of the drive system of the multi-objective optimization design scheme. Compared with the optimal design scheme with $\eta_{me}$ as a single objective, it is increased by 2% and 1.4%, respectively, and compared with the optimal design scheme with $\eta_{tr}$ as a single objective, it is improved by 26.5% and 73.6%, respectively, and the best economy is achieved.

**7. Conclusions**

(1) This study proposes a multi-objective optimization method for the drive motor of tractors. The objective functions are established based on the heat loss of the drive motor and the total efficiency of the driving system. Considering the inherent characteristics of the motor and the characteristics of the tractor's working conditions, a mathematical model of constraints is established, and the NSGA-II algorithm is used to design the optimization problem-solving process.

(2) An optimization example is developed, in which the single-target optimization results of the drive motor's optimal energy efficiency and the optimal mechanical transmission efficiency of the transmission system are taken as the control group for conducting the tractor electric drive system bench control test.

(3) The test results show that for the tractor model in the optimization example, compared with the control group, the proposed multi-objective optimization method can make the overall tractor system efficiency the highest, and the maximum and rated values of the total efficiency $\eta_q$ of the drive system of the multi-objective optimization design scheme. Compared with the optimal design scheme with $\eta_{me}$ as a single objective, it is increased by 2% and 1.4%, respectively, and compared with the optimal design scheme with $\eta_{tr}$ as a single objective; it improved by 26.5% and 73.6%, respectively. It can provide an effective calculation method for the motor design problem in the subsequent development of the tractor electromechanical drive system.

In this paper, the multi-objective optimization design and experimental electric tractor drive motor verification are carried out. However, the continuous full load of electric tractors will also produce new requirements for the water tank volume, cooling flow, and cooling fan parameters of the hydraulic cooling system driving the motor. Compared with traditional diesel engines, the wider speed range of electric motors will inevitably produce complex cooling system efficiency optimization problems. Therefore, in the future, field tests will be carried out on electric tractor prototypes, or the optimal design method of the

drive motor cooling system will be studied according to the way of loading according to the dynamic load spectrum of the tractor.

**Author Contributions:** Conceptualization, M.L.; Data curation, M.L., Y.L., B.H. and S.L.; Formal analysis, Y.L., B.H. and S.L.; Funding acquisition, M.L. and L.X.; Investigation, L.X.; Software, M.L.; Supervision, L.X.; Validation, S.Z. and S.L.; Visualization, L.X.; Writing—original draft, M.L.; Writing—review & editing, S.Z. All authors have read and agreed to the published version of the manuscript.

**Funding:** The project is supported by Open project of the State Key Laboratory of Power System of Tractor: SKT2021002.

**Informed Consent Statement:** Informed consent was obtained from all subjects involved in the study.

**Data Availability Statement:** No new data were created or analyzed in this study. Data sharing is not applicable to this article.

**Conflicts of Interest:** The authors declare no conflict of interest.

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
