# Peer review of "Multi-Objective Optimization and Test of a Tractor Drive Motor"

_wevj, doi:10.3390/wevj13020043_

Round 1
Reviewer 1 Report
This is the comments on the Manuscript Number: World Electric Vehicle Journal
Manuscript ID: wevj-1585841
Type of manuscript: Article
Title: Multi-objective Optimization and Test of Tractor Drive Motor
Authors: Mengnan Liu , Yanying Li , Sixia Zhao , Bing Han , Shenghui Lei , Liyou Xu
Special Issue: Design, Analysis and Optimization of Electrical Machines and Drives for Electric Vehicles
Journal WEVJ (ISSN 2032-6653)
Rate the Manuscript:
- Significance to field and specialization of “World Electric Vehicle Journal” journal: good.
The paper contains the theoretical modeling of the electric drive tractor has the characteristics of continuous stepless speed, high operating efficiency, and ability to output electric power for modern agricultural facilities. The design and power matching of the drive motor have a great impact on the energy efficiency of the system. In this work, the inherent electromagnetic principle of the motor and the tractor’s operating characteristics, a 2DOF optimization system mathematical model with the lowest heat loss, and the highest energy efficiency of the drive system is established by analyzing the electromechanical system. Constraint boundaries are set in terms of drive motor structure, field weakening speed regulation capability, the requirements of tractor torque and power, etc. On the basis of non-dominated sorting genetic algorithm II, the solution process of a tractor drive motor optimization system is designed, and the example is optimized. Bench test verification was performed by using the design results of other commonly used vehicle drive motor optimization methods as a control group. Test results show that the proposed multi-objective optimization method can make the electric drive tractor achieve the highest system efficiency under frequent loading conditions, thereby proving the feasibility of the proposed optimization model and solution method. Moreover, this research can provide theoretical and technical references for the drive motor’s design and selection in the subsequent development process of electric drive tractors.
This study proposes a multi-objective optimization method for the drive motor of tractors. The objective functions are established based on the heat loss of the drive motor and the total efficiency of the driving system. Considering the inherent characteristics of the motor and the characteristics of the tractor’s working conditions, a mathematical model of constraints is established, and the NSGA-â…¡ algorithm is used to design the optimization problem solving process. An optimization example is developed, in which the single-target optimization results of the drive motor’s optimal energy efficiency and the optimal mechanical transmission efficiency of the transmission system are taken as the control group for conducting the tractor electric drive system bench control test. The test results show that for the tractor models in the optimization example, the proposed multi-objective optimization method can maximize the efficiency of the entire tractor system compared with the control group, thereby verifying the feasibility of the multi-objective optimization method. It can provide an effective calculation method for subsequent motor design problems in the development of tractor electric drive systems. In this paper, the multi-objective optimization design and experimental verification of electric tractor drive motor are carried out. However, the continuous full load of electric tractors will also produce new requirements for the water tank volume, cooling flow, and cooling fan parameters of the hydraulic cooling system driving the motor. Compared with traditional diesel engines, the wider speed range of electric motors will inevitably produce complex cooling system efficiency optimization problems. Therefore, in the future, field tests will be carried out on electric tractor prototypes, or the optimal design method of the drive motor cooling system will be studied according to the way of loading according to the dynamic load spectrum of the tractor
- Scientific content: good.
- Originality: good.
- Clarity and presentation: acceptable.
- Appropriateness for Journal: appropriate subject mater for the “World Electric Vehicle Journal”
- Need for rapid publication: no
- Recommendations: to sent after revision to World Electric Vehicle Journal.
Remarks
- The abstract is not concise. Please, rewrite it. I has compared abstract and Conclusions.
- Much more sholud evidence is need before achive the conclusions.
- Authors should make sure that they written every sentence to convey their meaning dearly to the not conclusion.
- Article should be written in an organized way.
- The manuscript should be checked by native speaker for correct grammar and spelling(for example: Ref.21- please correct).
- To improve the Referenses list authors can consider some of the papers from the review: Hydrogen Containing Nanofluids in the Spark Engine’s Cylinder Head Cooling System. Energies 2022, 15(1), 59, 20 p. https://doi.org/10.3390/en15010059
Reviewer 2 Report
See the attached.

Reviewer 3 Report
The paper entitled "Multi-objective optimization and design of tractor trailer systems" by Liu, Mengnan; Zhou, Zhili; Xu, Liyou; Zhao, Jinghui; Yan, Xianghai in Transactions of the Chinese Society of Agricultural Engineering, Volume 33, Number 8, 1 April 2017, pp. 62-68(7) must be cited.
Reviewer 4 Report
This manuscript puts forward a new method for the optimization of the driving system of electric tractor, and the experiment verifies the energy-saving effect of the tractor optimized by this method. The proposed optimization model is innovative. The main deficiencies are:
1. In the introduction, the configuration of electric tractor should be discussed, and relative references can be supplemented.
2. In the test part, more details (such as core parameters) of the main tests instruments need to be described.
3. The conclusion is not clear enough, and some of that need a quantitative summary.
